# The *Arabidopsis* homolog of human G3BP1 is a key regulator of stomatal and apoplastic immunity

Aala A Abulfaraj[1,2] , Kiruthiga Mariappan[1] , Jean Bigeard[3,4], Prabhu Manickam[1], Ikram Blilou[1], Xiujie Guo[1], Salim Al-Babili[1] , Delphine Pflieger[5,6] , Heribert Hirt[1,3,4] , Naganand Rayapuram[1]

Mammalian Ras-GTPase–activating protein SH3-domain–binding proteins (G3BPs) are a highly conserved family of RNA-binding proteins that link kinase receptor-mediated signaling to RNA metabolism. Mammalian G3BP1 is a multifunctional protein that functions in viral immunity. Here, we show that the *Arabidopsis thaliana* homolog of human G3BP1 negatively regulates plant immunity. *Arabidopsis g3bp1* mutants showed enhanced resistance to the virulent bacterial pathogen *Pseudomonas syringae* pv. *tomato*. Pathogen resistance was mediated in *Atg3bp1* mutants by altered stomatal and apoplastic immunity. *Atg3bp1* mutants restricted pathogen entry into stomates showing insensitivity to bacterial coronatine–mediated stomatal reopening. AtG3BP1 was identified as a negative regulator of defense responses, which correlated with moderate up-regulation of salicylic acid biosynthesis and signaling without growth penalty.

## Introduction

RNA-binding proteins (RBPs) play a critical role in the regulation of gene expression, a feature that relies on their conformational plasticity and their capacity to interact with distinct targets (Jonas & Izaurralde, 2013; Castello et al, 2016). In mammals, Ras-GTPase–activating protein SH3-domain–binding protein 1 (G3BP1) belongs to a family of RBPs (Tourriere et al, 2001) that is located in stress granules (SGs) and contributes to their assembly (Tourriere et al, 2003). SGs are cytoplasmic aggregates containing stalled preinitiation complexes, which are thought to serve as sites of mRNA storage during the cell stress response (Anderson & Kedersha, 2008). Cells lacking G3BP1 are compromised to form SGs in response to eukaryotic initiation factor 2α phosphorylation or eIF4A inhibition but are SG competent upon heat or osmotic stress. G3BP1 interacts with 40S ribosomal subunits through the RGG motif, which is also required for G3BP-mediated SG formation

(Kedersha et al, 2016). G3BP1 is also involved in virus multiplication and modulates initiation of translation (Katsafanas & Moss, 2007; White et al, 2007; Panas et al, 2012). G3BP1 is a multifunctional protein that is highly conserved in all eukaryotes, but so far, only one protein AtG3BP-like protein (At5g43960) was identified as an *Arabidopsis thaliana* G3BP-like protein that localizes to plant SGs and plays a role in the *Arabidopsis* virus resistance (Krapp et al, 2017). Here, we identify the *Arabidopsis* G3BP homolog that belongs to a family of eight proteins and functions in plant immunity.

As in mammals, plants protect themselves against a wide variety of invading pathogens with different resistance strategies. This is accomplished by a sophisticated multilayered plant innate immune system (Dangl & Jones, 2001). Plants possess integrated immune signaling networks to facilitate rapid defense responses upon pathogen recognition, which in turn, restrict the further growth and spread of pathogens. In *Arabidopsis*, pathogen attack or treatment with a microbe-associated molecular pattern (MAMP), such as flg22 (the conserved 22–amino acid peptide derived from *Pseudomonas aeruginosa* flagellin), leads to the recognition of the MAMP by pattern recognition receptors that activate MAMP-triggered immunity (MTI). Receptor-mediated recognition of MAMPs leads to the induction of a battery of defense responses that include production of reactive oxygen species (ROS), an increase in intracellular calcium concentration (Yu et al, 2017), and activation of several *Arabidopsis* mitogen-activated protein kinases (MAPK), including MPK3, MPK4, MPK6, and MPK11 (Zipfel et al, 2006; Ranf et al, 2011; Frei dit Frey et al, 2014). These early events after MAMP perception in turn lead to intermediate defense responses mediated by transcriptional reprogramming of defense-related genes (Zipfel et al, 2004); production of antimicrobial compounds, including phytoalexins such as camalexin; callose deposition; biosynthesis of stress-related hormones such as salicylic acid (SA), jasmonic acid, and ethylene (Yu et al, 2017); and production of pathogen-related proteins such as PR1 (van Loon et al, 2006).

Recognition of MAMPs by pattern recognition receptors also triggers stomatal closure (Melotto et al, 2006; Singh & Zimmerli,

---

[1]Desert Agriculture Initiative, King Abdullah University of Science and Technology, Thuwal, Saudi Arabia    [2]Department of Biology, Science and Arts College, Rabigh Campus, King Abdulaziz University, Jeddah, Saudi Arabia    [3]Institute of Plant Sciences Paris-Saclay, CNRS, INRA, Université Paris-Sud, Université Evry, Université Paris-Saclay, Orsay, France    [4]Institute of Plant Sciences Paris-Saclay, Paris Diderot, Sorbonne Paris-Cité, Orsay, France    [5]Université Grenoble Alpes, CEA, Inserm, BIG-BGE, Grenoble, France    [6]CNRS, BIG-BGE FR3425, Grenoble, France

Correspondence: heribert.hirt@kaust.edu.sa

2013). Stomata are natural openings in the plant that are used by microbial pathogens for entry into the plant. However, these organs are essential for plant survival as they allow gaseous exchange and transpiration. Stomata are critical during both biotic and abiotic stress responses (Acharya & Assmann, 2009). The plant innate immune system limits infection by preventing the invasion of the apoplastic space and by compromising the proliferation of the pathogen postinvasion. *Pseudomonas syringae* secretes the phytotoxin coronatine (COR) that interferes with plant stomatal immunity (Melotto et al, 2006). In addition to disabling stomatal immunity, COR also inhibits apoplastic immunity through suppression of SA-mediated defenses (Zheng et al, 2012; Geng et al, 2014). SA is one of the key regulators of stomatal opening and closing. SA is a defense signal molecule against biotrophic and hemibiotrophic pathogens and regulates plant immune responses against several pathogens (Love et al, 2007; Chen et al, 2009; Wang et al, 2013; Ding et al, 2014). MAMPs also have been shown to induce SA accumulation (Mishina & Zeier, 2007; Tsuda et al, 2008). SA synthesis and signaling pathways are necessary for bacterial and MAMP-induced stomatal closure in *Arabidopsis* (Melotto et al, 2006; Zeng et al, 2010). SA enhances defense signaling that leads to transcriptional reprogramming (Moore et al, 2011) and ROS accumulation (Sato et al, 2010; Xu et al, 2014). Furthermore, *Arabidopsis* mutants such as *acd* (accelerated cell death) and *ssi* (suppressor of salicylate insensitivity of *npr1-5*), which express high levels of SA, are shown to be more resistant to pathogen infection and constitutively express defense genes including *PR* genes. In contrast, mutants with impaired SA biosynthesis such as *sid2* (salicylic acid induction-deficient-2) and *eds5* (enhanced disease susceptibility-5) enhance disease susceptibility to various pathogens (Ding et al, 2014). Mutations in SA accumulation or signaling genes and *Arabidopsis* transgenic plants with the bacterial gene *nahG* that encodes an SA hydrolase were compromised in defense against *P. syringae* pv. *tabaci*, *Phytophthora parasitica*, or *Cercospora nicotianae* (Delaney et al, 1994).

In this study, we characterize G3BP1 that belongs to a family of eight proteins in *Arabidopsis*. By studying loss-of-function mutants, we show that G3BP1 plants are resistant to *Pst* infection, have higher levels of SA, and exhibit closed stomata and constitutive defense gene expression. In addition, overexpressor lines exhibited stomatal opening as in the WT but were more susceptible to *P. syringae* infection. Our results establish that AtG3BP1 negatively regulates plant stomatal and apoplastic immunity via SA-mediated defense.

# Results

### AtG3BP1 is a homolog of the human HsG3BP1

Human G3BP1 protein contains two well-characterized domains, namely the nuclear transport factor 2 domain and an RNA recognition motif. We searched the TAIR10 database for all proteins with these two conserved domains. The *Arabidopsis* genome codes for eight proteins which satisfy these criteria. The eight *Arabidopsis* G3BPs are named AtG3BP1 to AtG3BP8 as indicated in Fig S1A. The independent identification of different members of the AtG3BP family by several groups and the use of different naming systems

have presented conflicts in the available literature, and a unified classification of AtG3BPs is desirable for further studies. A multiple alignment and phylogenetic tree for the relationship of G3BPs in *Arabidopsis* and human was generated using Molecular Evolutionary Genetics Analysis Version 7 (MEGA7) based on the amino acid sequences (Fig 1A) (Kumar et al, 2016). The phylogenetic tree was reconstructed using the neighbor-joining method (Saitou & Nei, 1987). Spliced isoforms from a single gene were eliminated in the phylogenetic tree for simplicity. AtG3BP1 is a homolog to human HsG3BP1 with an amino acid identity of 32% (Fig S1B).

### *Atg3bp1* mutant lines show a normal developmental phenotype

Two T-DNA insertion lines in the gene AT5G48650 were obtained from the National Arabidopsis Stock Center (NASC), *Atg3bp1-1* (SAIL_1153_H01) and *Atg3bp1-2* (SALK_027468). Using allele-specific primers, homozygous *Atg3bp1-1* and *Atg3bp1-2* mutants were identified and the site of insertion was determined by sequencing. In *Atg3bp1-1*, the T-DNA was inserted in the 5$^{th}$ exon 844 bp downstream of the ATG start site. In *Atg3bp1-2*, the T-DNA was inserted in the 9$^{th}$ exon 1,280 bp downstream of the ATG start site (Fig 1C). Determination of the relative transcript levels by qPCR indicated that both *Atg3bp1-1* and *Atg3bp1-2* were loss-of-function mutants (Fig 1D). For overexpression studies, a cauliflower mosaic virus *35S promoter*–At*G3BP1*–GFP construct was introduced into Col-0 WT background to give homozygous overexpression lines OE2 and OE3. OE2 and OE3 showed higher expression levels of *AtG3BP1* transcripts compared with WT controls (Fig 1E). Neither the loss-of-function mutants nor the overexpression lines showed visible growth or developmental phenotypes (Fig 1F).

### AtG3BP1 is expressed and localized to different cellular compartments

To study the in planta expression of *AtG3BP1*, stably transformed promoter–GUS (β-glucuronidase) reporter lines were produced. GUS analysis was performed on seedlings of 8–14 d and staining was carried out from 1 to 12 h. *AtG3BP1* was expressed in all tissues under these conditions (Fig 2A).

We then assessed the subcellular localization using several in silico prediction tools, where AtG3BP1 was predicted to potentially sort to the chloroplast and nucleus (Fig S2). To validate these predictions, we expressed an AtG3BP1–GFP fusion protein under the cauliflower mosaic virus 35S promoter in *Arabidopsis* stable lines (OE2) (Fig 2B). Confocal images indicated that AtG3BP1 is localized to various compartments within the cytoplasm (Fig 2C). We see expression of the recombinant protein in the cytoplasm of stomata in the hypocotyl region and true leaves (Fig 2C, ii–iv).

We also examined the subcellular localization of GFP-tagged AtG3BP1 transiently expressed in *Nicotiana benthamiana* leaf epidermal cells. Coinfiltration of vectors expressing AtG3BP1–GFP or Serrate–CFP, which served as internal reference marker for the nuclear compartment, showed that AtG3BP1–GFP was localized to the cytoplasm (Fig 2D).

### Loss of *AtG3BP1* alters classical MTI responses

To evaluate the role of *AtG3BP1* in MTI, *Atg3bp1* mutant lines were challenged with *P. syringae* pv. *tomato* DC3000 *hrcC*$^-$ (*Pst hrcC*$^-$) that

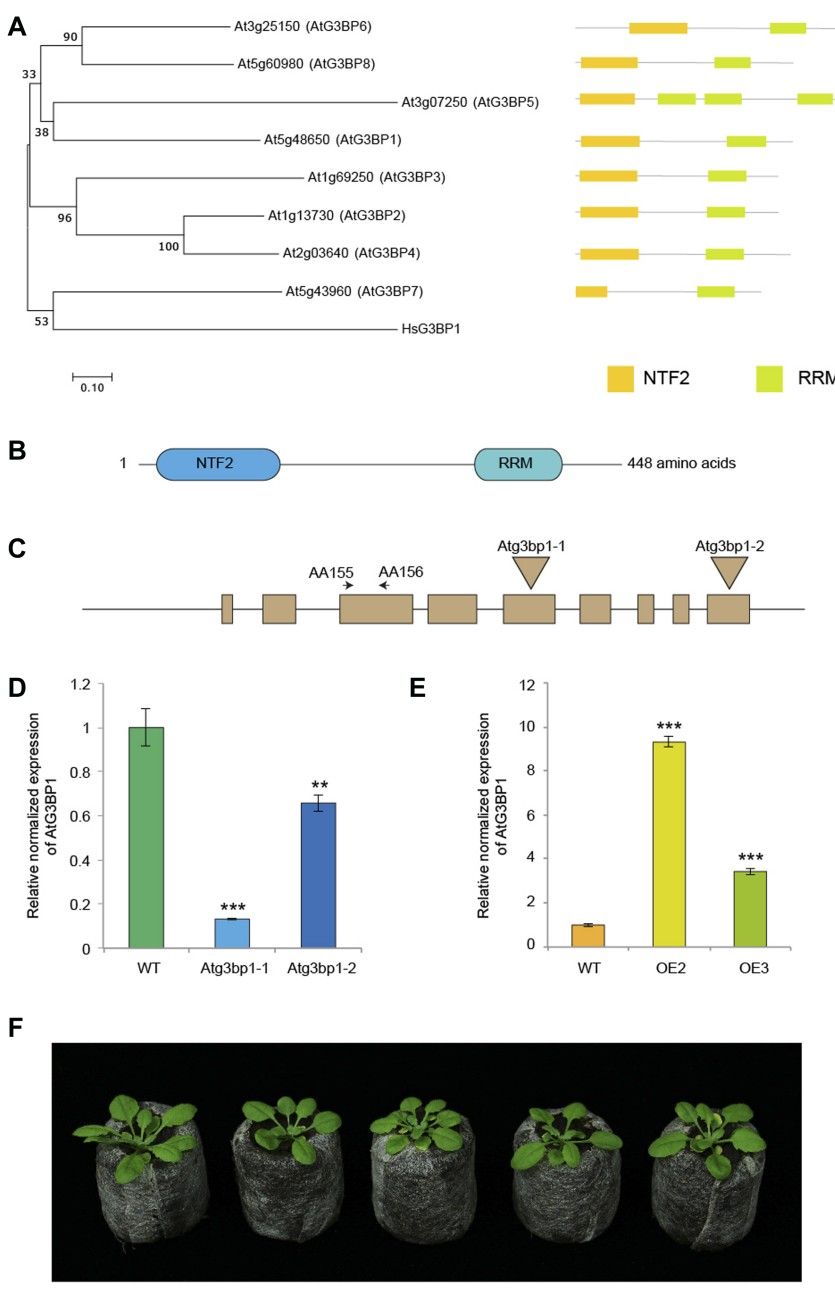

**Figure 1. AtG3BP1 is a homolog of the human HsG3BP1.**

**(A)** Neighbor-joining phylogenetic tree of the AtG3BPs superfamily and HsG3BP1 and Schematic comparison of the domain architecture of the AtG3BPs superfamily. The following domains are indicated: nuclear transport factor 2 (NTF2) and RNA recognition motif (RRM). The optimal tree with the sum of branch length = 5.30337989 is shown. The percentage of replicate trees in which the associated taxa clustered together in the bootstrap test (500 replicates) is shown next to the branches (Felsenstein, 1985). The tree is drawn to scale, with branch lengths in the same units as those of the evolutionary distances used to infer the phylogenetic tree. Evolutionary analyses were conducted in MEGA7 (Kumar et al, 2016). The evolutionary distances were computed using the Poisson correction method and are in the units of the number of amino acid substitutions per site. The analysis involved nine amino acid sequences. All positions containing gaps and missing data were eliminated. **(B)** AtG3BP1 protein structure showing the relative position of NTF2 and RRM domains. **(C)** The intron/exon structure of AT5G48650 and T-DNA insertional mutation sites in *Atg3bp1-1* and *Atg3bp1-2* mutant lines. Filled box represent exons, and AA155 and AA156 denote the positions of the primers used in the gene expression analysis. **(D)** *Atg3bp1-1* and *Atg3bp1-2* are loss-of-function mutants: expression levels of *Atg3bp1-1* and *Atg3bp1-2* by qPCR relatively to Col-0 WT (set at 1) with primers spanning the third exon (AA155 and AA156). *UBQ10* and *actin* expression levels were used for normalization. **(E)** Expression levels of *AtG3BP1* OE2 and OE3 by qPCR relatively to Col-0 WT (set at 1) with primers spanning the 3$^{rd}$ exon (AA155 and AA156). *UBQ10* and *actin* expression levels were used for normalization. **(F)**. Morphological phenotype of WT (Col.0), At*g3bp1-1*, and At*g3bp1-2* mutants, and *35S::AtG3BP1–GFP* (OE2) and *35S::AtG3BP1–GFP* (OE3) transgenic lines. 4-week-old jiffy peat pellets–grown plants are shown.

lacks the type III secretion system. *Arabidopsis* At*g3bp1-1* and At*g3bp1-2* mutants had reduced bacterial titers 3 hpi and 72 hpi compared with WT plants (Fig 3A). Conversely, the *AtG3BP1* over-expressor lines OE2 and OE3 harbored higher bacterial titers 72 hpi compared with WT plants (Fig 3B). Taken together, these data suggest that *AtG3BP1* negatively regulates disease resistance to *Pst hrcC⁻*. In addition, the difference in bacterial titers in the At*g3bp1-1* and At*g3bp1-2* mutants after 3 hpi suggests that *AtG3BP1* regulates the entry of *Pst hrcC⁻* into the plant.

Because *Arabidopsis* resistance to *Pst hrcC⁻* was increased in both *Atg3bp1* mutant alleles *Atg3bp1-1* and *Atg3bp1-2* (Fig 3A), we looked for cellular responses involved in MTI that were deregulated.

We first evaluated an early MTI response, the production of ROS (Kadota et al, 2014) in both alleles and observed a stronger ROS burst in At*g3bp1* mutant lines compared with WT plants after treatment with flg22 (Fig 3C).

Because MTI has been reported to be associated with transcriptional reprogramming (Navarro et al, 2004; Zipfel et al, 2004), we investigated the role of *AtG3BP1* in MAMP-triggered transcriptional responses. Quantitative RT-PCR (qRT-PCR) analysis showed that the classical MTI markers (Yang & Klessig, 1996; Asai et al, 2002) *FRK1*, *PR1*, and *WRKY29* were significantly more induced in untreated *Atg3bp1-1* when compared with WT plants. Besides, flg22 treatment further induces the expression of *FRK1* and *PR1* in the mutants

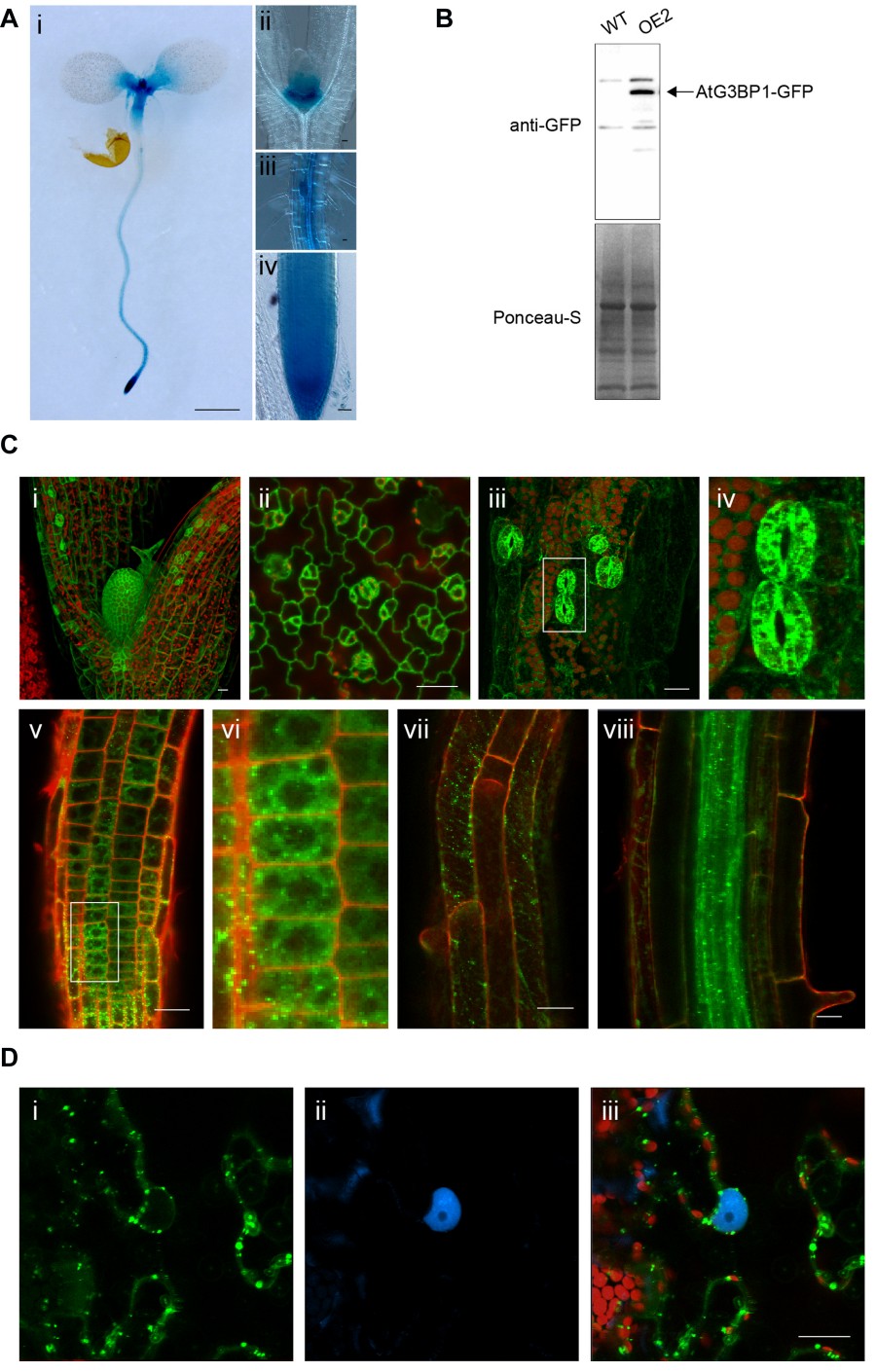

**Figure 2. AtG3BP1 is expressed and localized to different cellular compartments.**
**(A)** Expression patterns for the *GUS* reporter gene under the control of the *AtG3BP1* promoter in transgenic *Arabidopsis* seedlings. Histochemical GUS staining at the seedling stage. (i) 8-d-old seedling stained for 2 h. (ii) Shoot apical meristem of 14-d-old seedling stained for 12 h. (iii) Lateral root of 14-d-old seedling stained for 8 h. (iv) Root apical meristem of 12-d-old seedlings stained for 2 h. Scale bar: (i) = 1,000 μm, (ii–iv) = 20 μm. **(B)** Immunoblot of WT, OE2 plants expressing *35S::AtG3BP1–GFP* probed with anti-GFP antibody (upper panel) and protein loading control with Ponceau S staining (lower panel). **(C)** AtG3BP1 expression and protein localization in *Arabidopsis* roots (8 d) and leaves (12 d). Stable transgenic plants overexpressing AtG3BP1–GFP (OE2) reveal that AtG3BP1 is mainly localized to various cytoplasmic compartments. Confocal laser scanning microscopy images of GFP fluorescence (green) and propidium iodide (PI) fluorescence (red) of *Arabidopsis* seedlings in the (i) shoot apical meristem, (ii) leaf epidermis, and (iii) hypocotyl. Panel (iv) is the enlargement of boxed area in (iii), showing details of stomata localization. (v) Root apical meristem–epidermal layer. Panel (vi) is the enlargement of boxed area in (v), showing details of localization of AtG3BP1 to cytoplasmic compartments and structures. The protein is excluded from the nucleus. (vii, viii) Root differentiation zone. Scale bar: 20 μm. **(D)** Subcellular localization of transiently expressed *35S::AtG3BP1–GFP* and *pUBi::Serrate–CFP* in *N. benthamiana* cells. The indicated combinations of fluorescently tagged proteins were transiently expressed in 4-week-old *N. benthamiana* leaf epidermal cells. The localization was visualized 72 h after infiltration by laser scanning confocal microscopy. GFP fluorescence is in green and CFP fluorescence is in blue. Scale bars = 20 μm.

compared with WT plants. Although the expression of *WRKY29* and *MYB51* was higher in the mutants compared with the WT after flg22 treatment, this was statistically not significant (Fig 3D).

In plants, ROS act as signaling molecules and have been shown to play a role in multiple stress responses (Dalton et al, 1999). ROS can also induce a variety of auto-oxidative chain reactions leading to signaling events and ultimately to the destruction of organelles and macromolecules (Mittler et al, 2004; O'Brien et al, 2012). To

further clarify the role of *AtG3BP1* in stress responses, WT plants and *Atg3bp1* mutants were specifically stained for superoxide radicals using the nitroblue tetrazolium (NBT) and for hydrogen peroxide using 3,3′-DAB. Polymerization of DAB and NBT can be detected as a brown precipitate in the presence of hydrogen peroxide and as a blue precipitate in the presence of superoxide, respectively. Compared with WT plants, a strong DAB and NBT stain appeared in *Atg3bp1-1* and *Atg3bp1-2* mutants (Fig 3E and F).

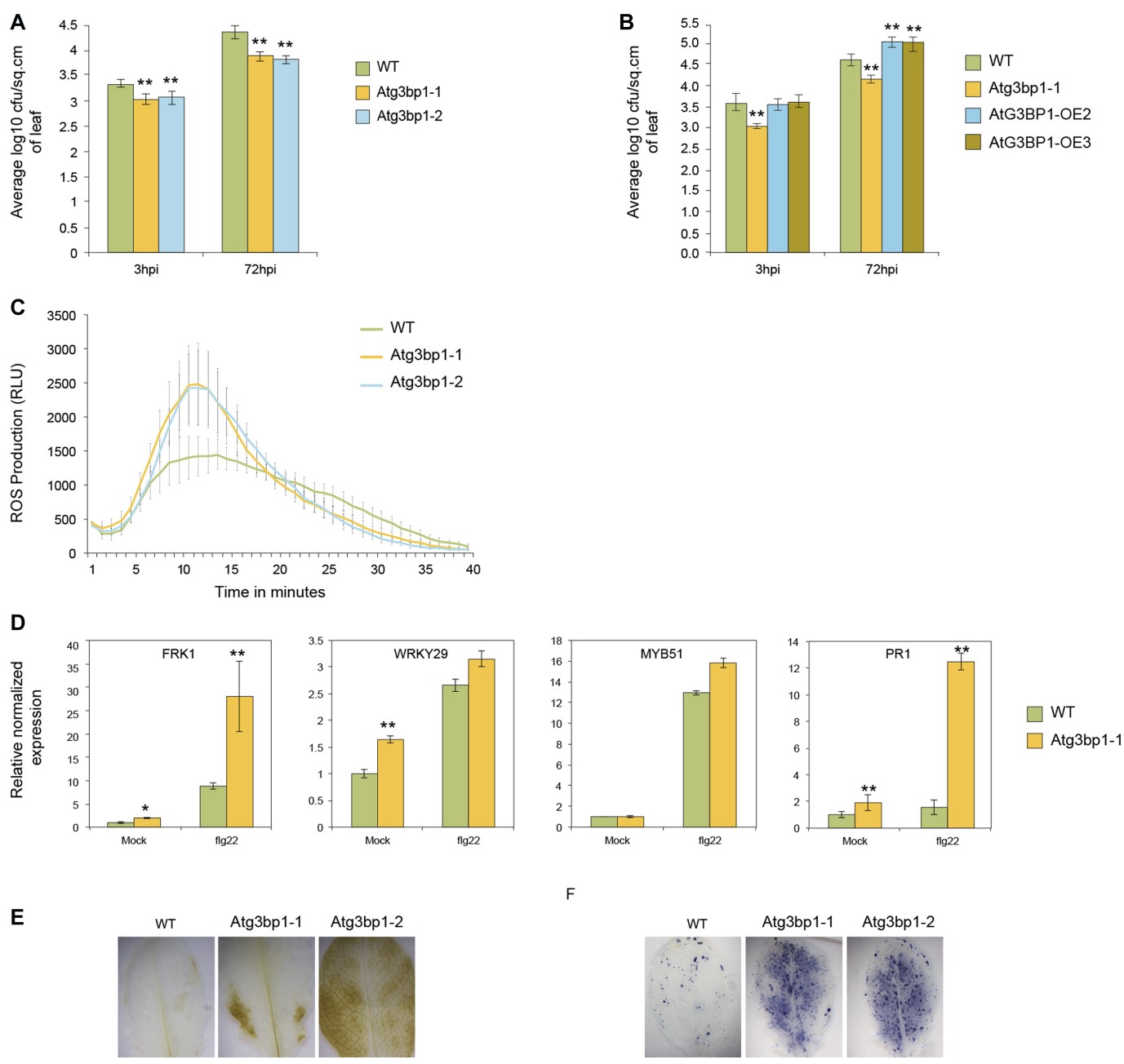

**Figure 3. The Atg3bp1 mutants display enhanced disease resistance and alter classical MTI responses.**
**(A)** Bacterial titers evaluated 3 and 72 h after spray inoculation with 1 × 10⁸ CFU·ml⁻¹ *Pst hrcC⁻* in WT and two *Atg3bp1* mutant lines (*Atg3bp1-1* and *Atg3bp1-2*). Values represent SEM from three independent experiments. Asterisks indicate significant differences from the WT as determined by Mann–Whitney *U* two-tailed test (*P* ≤ 0.01). **(B)** Bacterial titers evaluated 3 and 72 h after spray inoculation with 1 × 10⁸ CFU·ml⁻¹ *Pst hrcC⁻* in WT, *Atg3bp1-1*, and two At*G3BP1* overexpressing lines (OE2 and OE3). Values represent SEM from three independent experiments. Asterisks indicate significant differences from the WT as determined by Mann–Whitney *U* two-tailed test (*P* ≤ 0.01). **(C)** flg22-induced ROS burst in the WT and *Atg3bp1* mutant lines. Leaf discs from 4-week-old plants were treated with 1 μM flg22 over 40 min. The data were shown as means from 12 leaf discs. **(D)** flg22-induced marker gene expression in the WT and *Atg3bp1-1* plants. 14-d-old seedlings were treated with 1 μM flg22 for 1 h. Transcripts levels were determined by qRT-PCR relatively to Col-0 WT (set at 1). *UBQ10* and *actin* expression levels were used for normalization. **(E, F)** Detection of hydrogen peroxide accumulation by DAB staining (F) and of superoxide anion accumulation by NBT staining (G) in WT and *Atg3bp1* mutants.

### AtG3BP1 is a negative regulator of stomatal immunity

Plants restrict the invasion of bacteria after sensing the MAMPs through stomatal closure (Melotto et al, 2006; Zeng et al, 2010). We then hypothesized that the lower bacterial titers that we observed

3 hpi in At*g3bp1* mutant lines compared with WT after spray in-oculation (Fig 3A) might result from the prevention of bacterial entry into the leaves via the stomata. To test this hypothesis, we examined the stomatal aperture in At*g3bp1-1* mutants. Stomata of At*g3bp1-1* were more closed compared with those of WT plants

(Fig 4A), suggesting that diminished levels of *AtG3BP1* caused the stomatal response phenotype observed in *Atg3bp1-1* and *Atg3bp1-2*.

We then treated At*g3bp1-1* mutant lines with flg22 or abscisic acid (ABA), both of which are known to promote stomatal closure. After 1 h of ABA treatment, the stomata in both WT and *Atg3bp1-1*

Q:9

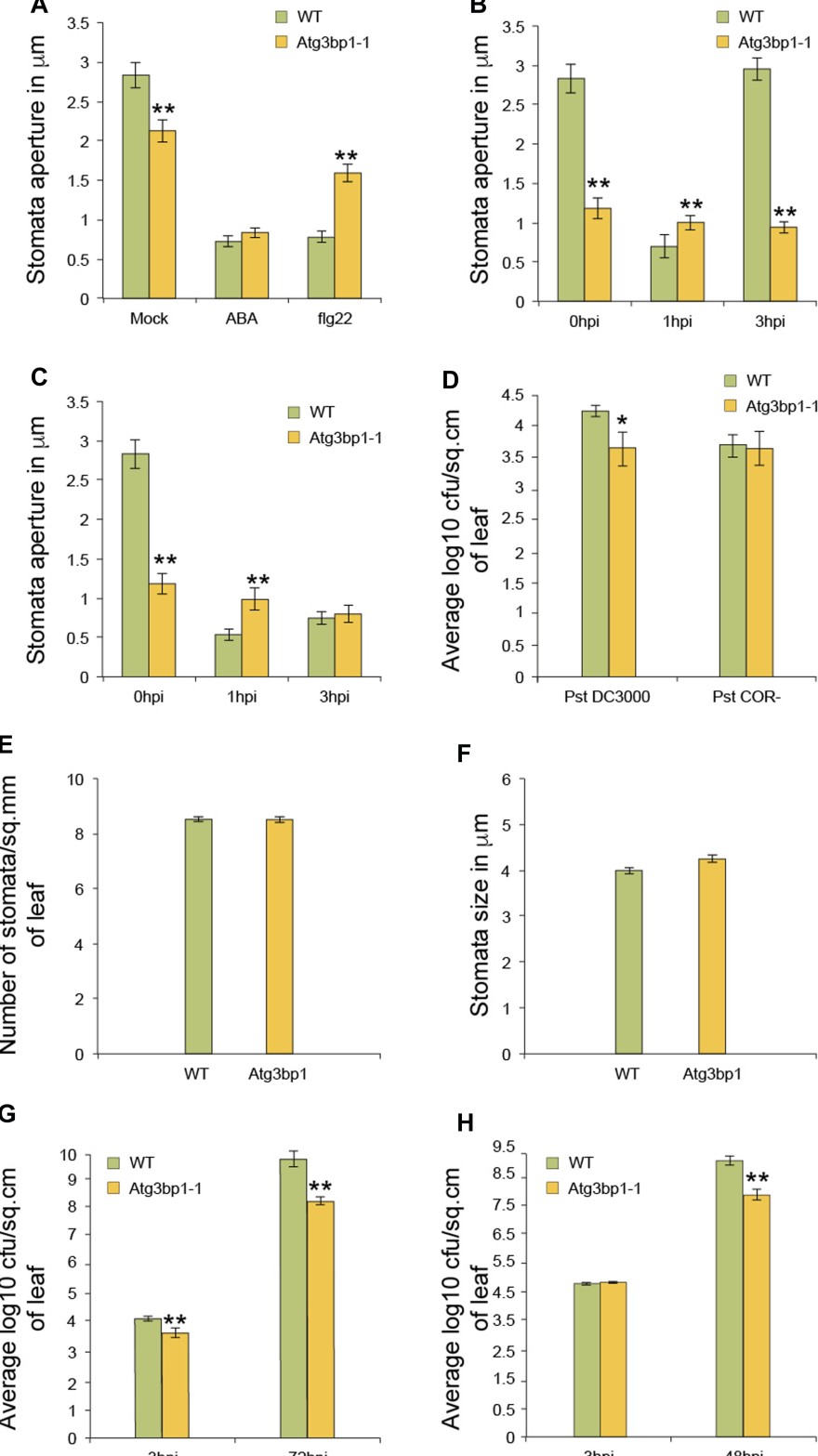

**Figure 4. *AtG3BP1* is associated with stomatal immunity.**
**(A)** Stomata closure upon ABA and flg22 treatment. Epidermal peels of 5-week-old WT and *Atg3bp1-1* plants were floated in stomata buffer with either 20 μM ABA or 1 μM flg22. Stomatal apertures were evaluated after 1 h. Values represent SEM from three independent experiments. Asterisks indicated significant differences from the WT plants determined by a Mann–Whitney *U* two-tailed test ($P \leq 0.01$). **(B, C)** Stomata closure upon *Pst* or *Pst COR⁻* inoculation. Epidermal peels of 5-week-old WT and *Atg3bp1-1* plants were floated in a suspension of $1 \times 10^8$ CFU·ml⁻¹ *Pst* or *Pst COR⁻* in stomata buffer. Stomatal apertures were evaluated at the indicated time points. Values represent SEM from three independent experiments. Asterisks indicate significant differences from the WT plants determined by a Mann–Whitney *U* two-tailed test ($P \leq 0.01$). **(D)** Bacterial titers evaluated 3 h after spray inoculation with $1 \times 10^8$ CFU·ml⁻¹ *Pst* or *Pst COR⁻* in WT and *Atg3bp1-1* plants. Values represent SEM from three independent experiments. Asterisks indicate significant differences from the WT plants as determined by a Mann–Whitney *U* two-tailed test ($P \leq 0.05$). **(E)** Stomata density in leaf abaxial surfaces of WT and Atg3bp1-1 (D, stomata per mm²). **(F)**. Average stomatal size in leaf abaxial surfaces of WT and Atg3bp1-1. Mean pore depth (l mm), which is assumed to be equivalent to guard cell width. **(G)** Bacterial titers evaluated 3 and 72 h after spray inoculation with $1 \times 10^8$ CFU·ml⁻¹ *Pst* in WT and *Atg3bp1-1*. Values represent SEM from three independent experiments. Asterisks indicate significant differences from the WT plants as determined by a Mann–Whitney *U* two-tailed test ($P \leq 0.01$). **(H)** Bacterial titers evaluated 3 and 48 h after infiltration inoculation with $1 \times 10^8$ CFU·ml⁻¹ *Pst* in WT and *Atg3bp1-1* plants. Values represent SEM from three independent experiments. Asterisks indicate significant differences from the WT plants as determined by a Mann–Whitney *U* two-tailed test ($P \leq 0.01$).

plants closed to the same extent, whereas the stomatal closure in *Atg3bp1-1* mutant lines was compromised in response to MAMP treatment (Fig 4A). This observation shows that although the ABA-dependent stomatal closure is not affected in the *Atg3bp1-1* mutants, the flg22-dependent pathway is compromised, confirming a role for *AtG3BP1* in the stomatal immune pathway.

*Arabidopsis* plants close stomata when they sense *Pst* within 1 hpi. However, *Pst* and other virulent bacteria induce the stomata to reopen via secretion of the chemical effector COR. Thus, WT *Arabidopsis* are found to be resistant to COR-deficient *Pst* mutants of *Pst COR⁻* (Melotto et al, 2006; Zeng et al, 2010). To test the role of *AtG3BP1* in stomatal reopening, leaf pieces were incubated with *Pst* or *Pst COR⁻* bacteria. Stomatal apertures in leaf epidermal peels of WT and *Atg3bp1-1* plants were subsequently assessed after 1 and 3 h. In WT plants, the stomata rapidly closed after treatment with *Pst* and reopened to normal levels after 3 hpi (Fig 4B). However, when compared with WT plants, *Atg3bp1-1* mutants were compromised in stomatal closure in nontreated controls at 0 hpi and 1 h after treatment (Fig 4B and C at 0 and 1 hpi, respectively). Importantly, *Atg3bp1-1* plants were completely insensitive to *Pst*-mediated stomatal reopening at 3 hpi (Fig 4C, 3 hpi). These data indicate that *Atg3bp1-1* mutants counteract COR-dependent reopening of stomata and that resistance to *Pst hrcC⁻* in *Atg3bp1* mutant lines may, therefore, be due to the stomatal closure response.

To further evaluate the role of *AtG3BP1* in stomatal immunity, WT plants and *Atg3bp1-1* mutant lines were challenged with *Pst* and *Pst COR⁻* by spray inoculation. Bacterial titers were evaluated at 3 hpi. Bacterial titers were significantly reduced in *Atg3bp1-1* compared with WT plants when challenged with *Pst* (Fig 4D). However, no significant differences in the bacterial titers of *Atg3bp1-1* and WT were found when challenged with *Pst COR⁻* (Fig 4D). We also measured the stomatal density and size in *Atg3bp1-1* for developmental defects. However, *Atg3bp1-1* mutant and WT showed similar stomatal density and size (Fig 4E and F), indicating that the reduced bacterial titers in *Atg3bp1-1* were not the result of smaller stomatal size or density but due to differences in stomatal behavior.

We further tested the disease phenotype of *Atg3bp1* mutant lines with the virulent bacterial pathogen *Pst*. *Atg3bp1-1* developed less disease symptoms and had lower bacterial titers 3 and 72 hpi than WT controls when challenged with *Pst* by spray inoculation (Fig 4G). Next, we investigated whether *AtG3BP1* is critical for *Arabidopsis* resistance even when the stomatal barrier was breached by infiltration inoculation of *Pst*, which bypasses the first barrier of defense, namely, stomatal closure that restricts the passage of bacteria into the plant (Melotto et al, 2006; Zeng et al, 2010). However, *Atg3bp1-1* showed an enhanced resistance phenotype when challenged with *Pst* by infiltration inoculation (Fig 4H), indicating that *AtG3BP1* also negatively regulates postinvasive disease resistance in mesophyll cells.

### Global transcriptomic profile shows that *AtG3BP1* negatively regulates defense gene expression

To identify the *AtG3BP1*-dependent genes, we performed whole transcriptome analysis (RNA-seq) with 14-d-old seedlings of WT and *Atg3bp1-1* genotypes treated for 24 h with or without $1 \times 10^8$ CFU·ml⁻¹ *Pst hrcC⁻* by spray inoculation. Samples from three independent biological experiments were collected for RNA extraction and RNA-seq was carried out on Illumina HiSeq (Illumina Inc.). The data discussed in this publication have been deposited in NCBI's Gene Expression Omnibus (GEO) and are accessible through GEO Series accession number GSE107786. Differentially expressed genes (DEGs) were based on at least a twofold change in gene expression and the false discovery rate (FDR) value ($P ≤ 0.05$) as described in the Bioinformatics part of the Materials and Methods section (Fig S3A). All the down- or up-regulated genes were overlapped to obtain the unique down- or up-regulated genes in mock–mock and mock–*Pst hrcC⁻* treated conditions (Fig 5A).

Performing hierarchical clustering by combining all the 1937 unique genes that are differentially expressed (Fig 5A) revealed several clusters as depicted in Fig 5B. Focusing on three specific clusters of interest, cluster IV comprised 160 DEGs that were highly up-regulated in *Atg3bp1-1* upon *Pst hrcC⁻* treatment but not in WT (Fig 5C). Enrichment analysis of Gene Ontology categories of cluster IV revealed genes associated with response to stress, biotic stimulus, and defense (Fig 5C). Therefore, *Atg3bp1* mutants respond more strongly to microbial challenge than WT. A constitutive defense response in *Atg3bp1-1* was observed for cluster VI, with 261 DEGs, which were highly up-regulated already in the absence of *Pst hrcC⁻* treatment (Fig 5C). Another large set of genes, cluster VII with 597 DEGs, all related to ROS, hydrogen peroxide metabolism, oxidation–reduction process, peroxidase activities, and oxidoreductase activities, was also found to be constitutively up-regulated in untreated *Atg3bp1-1* plants (Fig 5C).

The expression of several immunity-related genes in *Atg3bp1* mutants, including *WAK1* (*AT1G21250*), *WRKY40* (*AT1G80840*), *WRKY8* (*AT5G46350*), and *AZI1* (*AT4G12470*), and defensin-like protein (*AT3G59930*) was confirmed by qRT-PCR analysis (Fig S3B). Taken together, the global gene expression data suggest that *AtG3BP1* plays a negative role in a large subset of immunity-related genes.

### *AtG3BP1* is a key negative regulator of genes involved in SA biosynthesis and signaling

The transcriptome data indicated that a large set of immunity-related genes are either constitutively or more highly induced in *Atg3bp1* mutants upon microbial challenge. Among these genes, a strong enrichment was observed for SA-related genes (Fig S3C and D). For this purpose, we evaluated the expression levels of several known SA-mediated defense genes in *Atg3bp1-1* and WT seedlings by qRT-PCR. We found that the basal transcript levels of the CaM-binding protein 60 g (*CBP60g*) and SAR deficient-1 (*SARD1*) were higher in *Atg3bp1-1* than those in WT. *CBP60g* and *SARD1* are members of the CaM-binding protein family that regulate the expression of the SA biosynthetic gene isochorismate synthase 1 (*ICS1*) (Zhang et al, 2010; Wang et al, 2011). The expression of *ICS1* was also significantly higher in *Atg3bp1-1* compared with WT (Fig 6A). For SA signaling-related genes, the expression levels of *NIM1 interacting 1* (*NIMIN1*), transcription factor *WRKY38*, and *PR2* were tested and found to be higher in *Atg3bp1-1* compared with WT (Fig 6B). Expression of some of the SA marker genes *PAD4*, *EDS1*, and *EDS5* were also elevated in *Atg3bp1-1* compared to WT (Fig 6C). The

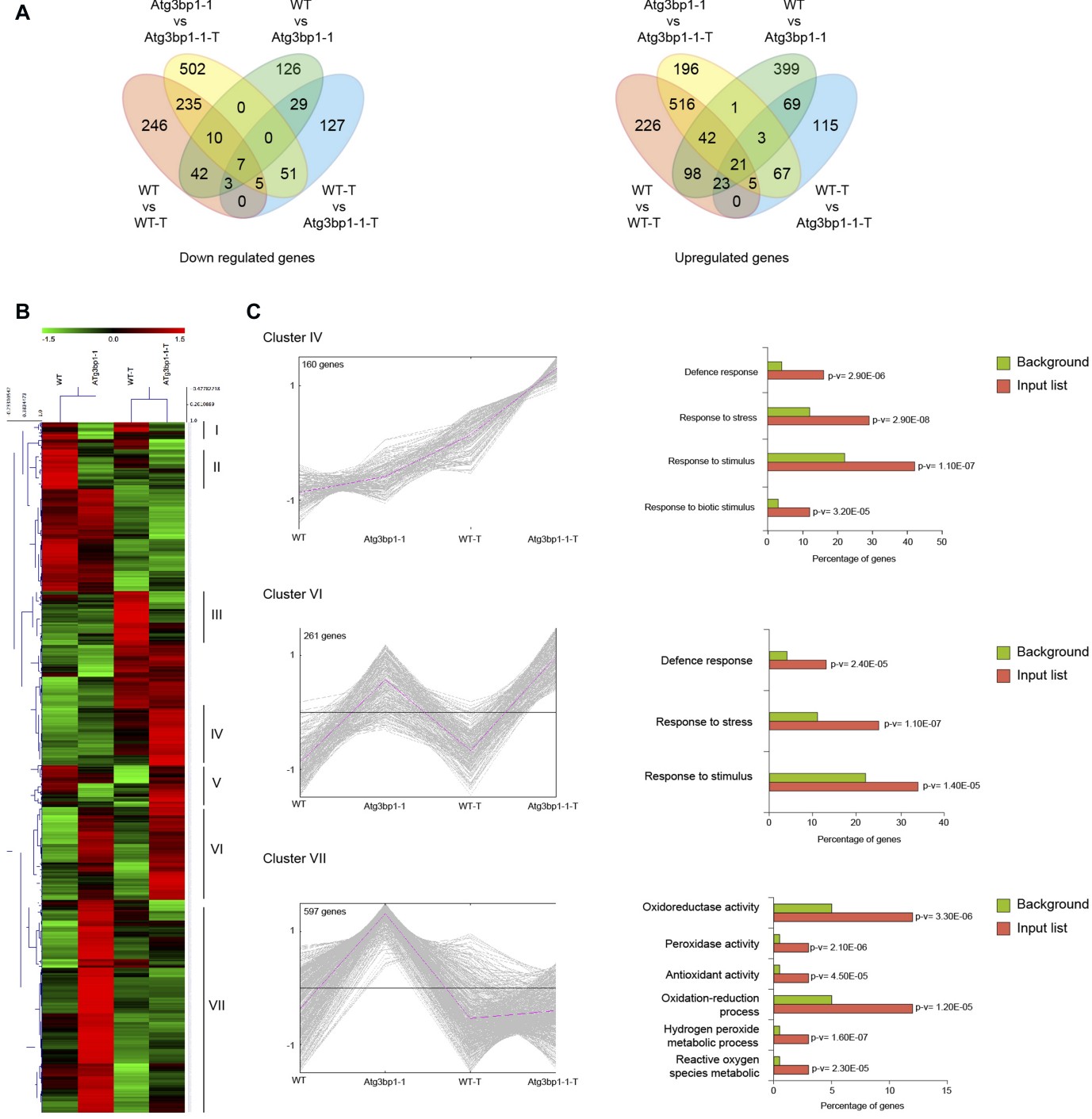

**Figure 5. Global transcriptomic profile shows that *AtG3BP1* is associated with defense responses.**

**(A)** Venn diagrams showing the number of the unique differentially expressed genes. The numbers in the Venn diagrams were obtained by overlapping the unique down- or up-regulated genes in mock–mock and mock-treated conditions (WT versus WT-treated, *Atg3bp1-1* versus *Atg3bp1-1*-treated, WT versus *Atg3bp1-1*, and WT-treated versus *Atg3bp1-1*-treated). **(B)** Heat map of *AtG3BP1*-induced genes with and without *Pst hrcC⁻* treatment in WT and *Atg3bp1-1* plants. The original fragments per kilobases million values were subjected to data adjustment by normalizing genes or rows and hierarchical clustering was generated with the average linkage method using MeV4.0. Red color indicates up-regulation and green indicates down-regulation. Clusters are defined in the text. **(C)** Expression profiles and enrichment of genes with GO terms for interesting clusters of *AtG3BP1*-induced genes with and without *Pst hrcC⁻* treatment in WT and *Atg3bp1-1* plants. Right panel: for each interesting cluster, enrichment in genes with GO terms related to immune response. The fold enrichment was calculated based on the frequency of genes annotated to the term compared with their frequency in the genome.

Q:10

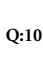

**Figure 6. Endogenous salicylic acid (SA) triggers stomatal closure and immune defense in Atg3bp1 mutants.**

**(A)** Expression of SA-biosynthesis–related genes in the WT and *Atg3bp1-1* 14-d-old seedlings. Transcript levels were determined by qRT-PCR relatively to Col-0 WT (set at 1). *UBQ10* and *actin* expression levels were used for normalization. **(B)** Expression of SA-signaling–related genes in the WT and *Atg3bp1-1* 14-d-old seedlings. Transcript levels were determined by qRT-PCR relatively to Col-0 WT (set at 1). *UBQ10* and *actin* expression levels were used for normalization. **(C)** Expression of SA-accumulation–related genes in the WT and *Atg3bp1-1* 14-d-old seedlings. Transcript levels were determined by qRT-PCR relatively to Col-0 WT (set at 1). *UBQ10* and *actin* expression levels were used for normalization. **(D)** Endogenous SA levels in 14-d-old seedlings of WT and *Atg3bp1-1* mutant plants. Values represent SEM from three independent experiments. **(E)** Stomatal aperture in epidermal peels of 5-week-old WT and *Atg3bp1-1* plants were floated in stomata buffer and treated 1 h with 100 μM SA followed by inoculation with either *Pst* or *Pst COR−*. Stomatal apertures were evaluated after 3 h. Values represent SEM from three independent experiments. Asterisks indicate significant differences from WT plants determined by a Mann–Whitney *U* two-tailed test ($P \leq 0.01$).

transcription factors *WRKY40* and *AZI1* (azelaic acid induced-1) are often used as markers for SA-dependent host defense responses (Lamb et al, 1989; Baker et al, 1997). *WRKY40* and *AZI1* were also up-regulated in *Atg3bp1-1* compared with WT controls (Fig S3B). Overall, the hyperinduction of SA defense-related genes correlates with the resistance to *Pst* observed in the *Atg3bp1* mutants.

### AtG3BP1 regulates stomatal immunity via SA signaling

We showed that *Arabidopsis G3BP1* is critical for stomatal immunity and that *AtG3BP1* is a negative regulator of SA biosynthesis and signaling gene expression. Since SA is required for innate immunity–mediated stomatal closure (Zeng et al, 2011b), we thus quantified endogenous SA levels in *Atg3bp1* mutant lines and found significant increase in SA levels in *Atg3bp1-1* mutants compared with WT seedlings (Fig 6D).

Because an increase in endogenous SA can trigger stomatal closure, we hypothesized that an increase in the expression of SA-related genes might trigger stomatal closure and in turn lead to immune resistance in *Atg3bp1* mutants. For this purpose, stomatal apertures were assessed in WT and *Atg3bp1-1* after a 1-h pre-treatment of SA followed by inoculation with either *Pst* or *Pst COR⁻*. The inoculation with *Pst* did not reopen the stomata in WT when pretreated with SA. But in WT leaves that had not been treated with SA, stomata completely reopened to normal sizes after 3 h of in-oculation with *Pst* because of COR produced by the pathogen. In WT leaves pretreated with SA followed by inoculation with *Pst COR⁻*, the stomata remained closed because of the lack of COR. In contrast, the stomata in *Atg3bp1-1* leaves remained closed and the treatment with either *Pst* or *Pst COR⁻* had no effect on the stomata (Fig 6E). Collectively, these results confirm that *AtG3BP1* functions in SA-mediated stomatal closure and reopening in response to bacterial pathogens.

## Discussion

In this study, we report the identification of the *Arabidopsis G3BP1* as an important player in SA-dependent stomatal immunity. AtG3BP1 is a member of a family of eight proteins in *Arabidopsis* that are orthologs to the human G3BP gene family. Mammalian G3BP1 is an evolutionarily highly conserved RBP that was identi-fied through its interaction with a Ras-GTPase–activating protein (Tourriere et al, 2003). G3BPs have been mainly studied in mammals where they are involved in many processes, such as the formation of SGs and the involvement in virus resistance (Tourriere et al, 2003; Katsafanas & Moss, 2007; Panas et al, 2012). However, so far, no G3BP has been functionally characterized in plant stomatal and apo-plastic immunity. Expression profiles of the different members of the family in *Arabidopsis* are subjected to changes according to developmental stages, tissue specificity, and environmental per-turbations. From these patterns, the genes might perform similar biochemical functions in a tissue- and developmental-specific manner and respond to different conditions (Fig S4). AtG3BP7 (At5g43960), formerly known as AtG3BP-like protein, was identified as an *Arabidopsis* G3BP-like protein that localizes to plant SGs and

plays a role in the *Arabidopsis* virus resistance. It has been shown that SG formation and function is conserved between mammalian and plant cells during stress. Moreover, plant viruses have the ability to bind to AtG3BP7, preventing the formation of SGs, as in mammals (Krapp et al, 2017).

The enhanced disease resistance to *Pst* observed in two in-dependent *Atg3bp1* loss-of-function mutant lines indicates a role for *AtG3BP1* in plant innate immunity. Several lines of evidence also showed that *AtG3BP1* is involved in MTI. *Atg3bp1* mutants have reduced sensitivity to *Pst hrcC⁻*, a nonvirulent pathogen used to study MTI. *Atg3bp1* mutants also showed enhanced ROS accumu-lation in response to flg22 and up-regulation of classical MTI marker genes and stomatal innate immunity.

*Atg3bp1-1* mutants showed the activation of defense responses leading to constitutive stomatal closure. More importantly, the striking insensitivity to COR-dependent stomatal reopening in *Atg3bp1* mutants explains the enhanced resistance phenotype of *Atg3bp1* mutants to *Pst* by spray inoculation. Moreover, in spite of overcoming the stomatal barrier by infiltration of *Pst*, the *Atg3bp1* mutants showed enhanced disease resistance. Taken together, these data indicate that *Arabidopsis G3BP1* is a regulator of sto-matal and apoplastic immunity.

The analysis of the SA-deficient *nahG* mutant and SA-biosynthetic *sid2/eds16* mutant revealed that SA biosynthesis is required for MAMP-induced stomatal closure (Melotto et al, 2006; Zeng et al, 2010). *Atg3bp1* mutant plants showed up-regulation of SA biosynthesis and signaling marker genes and accumulated SA. It was also shown that SA induces stomatal closure by peroxidase-mediated extracellular ROS production (Mori et al, 2001; Khokon et al, 2011). Consistent with a role of *AtG3BP1* in SA-induced ROS production, *Atg3bp1* mutants showed constitutive up-regulation of genes related to ROS production and enhanced DAB and NBT staining for hydrogen peroxide and superoxide, respectively. The SA-dependent pathway is also critical in mediating *Pst* resistance in the COR-insensitive *coi-20* mutants (Kloek et al, 2001). Thus, SA signaling pathway is required for stomatal immunity and apoplastic defenses against *Pst* (Kloek et al, 2001; Melotto et al, 2006; Zeng et al, 2010, 2011a). Our data support the idea that *Atg3bp1* immunity against *Pst* including stomatal defense is correlated with the ac-tivation of SA signaling.

Recently, COR was found to induce the NAC transcription factors (petunia NAM and *Arabidopsis* ATAF1, ATAF2, and CUC2) *ANAC019*, *ANAC055*, and *ANAC07* to repress *ICS1* and induce *BSMT1* to inhibit SA accumulation, thus mediating reopening of the stomata to facilitate bacterial entry (Zheng et al, 2012). Interestingly, in *Atg3bp1-1* mu-tants, two of these NAC TFs (*ANAC019* and *ANAC055*) were up-regulated in untreated plants (Fig S5) but nonetheless result in accumulation of SA, requiring another explanation for the in-sensitive phenotype to COR-induced stomatal reopening.

RBPs, especially those involved in response to stress, have been shown to regulate gene expression at different stages, from transcription to posttranslational levels (Mastrangelo et al, 2012). Posttranscriptional regulations are important processes in eu-karyotes that enable them to adapt to changes in their environ-ment. Human G3BP1 is located in SGs, which are sites of mRNA storage during stress responses that are involved in regulating mRNA stability and translatability (Anderson & Kedersha, 2008). In

humans, SGs are highly dynamic entities that recruit many signaling proteins in a transient manner until the cells adapt to stress or die. Hence, beyond their role as mRNA triage centers, SGs are proposed to constitute RNA-centric signaling hubs analogous to classical multiprotein signaling domains such as transmembrane receptor complexes which communicate a "state of emergency" (Kedersha et al, 2013). A good example for this as a signaling hub is the one shown by HsG3BP1 which functions in regulating double-stranded RNA-dependent protein kinase activation (Reineke et al, 2015). Given the highly conserved nature of eukaryotic G3BPs, it is tempting to speculate that plant G3BP1 might function in an analogous fashion during stress. This might also explain the fact that *Atg3bp1* mutants showed enhanced ROS accumulation in response to flg22, which would suggest that AtG3BP1 functions upstream of ROS production. In this context, it should also be noted that ROS production does not only act as an antimicrobial agent in MTI, but also as a secondary messenger that triggers downstream defense responses, including stomatal closure and up-regulation of MTI marker genes (Melotto et al, 2006; Kadota et al, 2014; Yu et al, 2017).

Interestingly, whereas we identified eight G3BPs in *Arabidopsis*, the human genome only encodes for two HsG3BPs. Expression analysis of the *Arabidopsis* genes shows, however, that these AtG3BPs are expressed in a highly tissue- and developmental-specific fashion (Fig S4), suggesting that the different genes might perform similar functions in different cellular contexts.

Collectively, this work reveals that plants and mammals possess a set of highly conserved G3BP proteins with a role in immunity and further investigations are warranted to unravel the roles of G3BPs in plants.

## Materials and Methods

### Biological materials and growth conditions

*Arabidopsis* ecotype Columbia-0 plants were used as WT in all experiments. Two T-DNA mutant lines *Atg3bp1-1* (SAIL_1153_H01) and *Atg3bp1-2* (SALK_027468) were obtained from the National Arabidopsis Stock Center (NASC) and genotyped by PCR amplification of insertion-specific or WT-specific fragments with primers listed in Table S1. Plants were grown on soil at 21°C with a 12-h photoperiod or as seedlings on sterile one-half–strength Murashige and Skoog (MS) medium with a 16- or 8-h photoperiod for 14 d.

Bacterial strains *P. syringae* pv. *tomato Pst* DC3000, *Pst* DC3000 *hrcC*, and *Pst* DC3000 *COR⁻* were cultured at 28°C from glycerol stock on NYGA agar plate (5 g/liter bactopeptone, 3 g/liter yeast extract, 20 ml/liter glycerol, and 15 g/liter agar) containing 50 mg/ml rifampicin.

### Cloning and generation of overexpressor plants

All the details are described in the Materials and Methods section of the Supplemental Information.

### Various treatments, pathogen infection, and MTI assays

MAMP and hormone treatments, bacterial pathogen infection assays, stomatal aperture assay, oxidative burst assay, superoxide radical and hydrogen peroxide staining, and SA quantification are described in detail in the Materials and Methods section of the Supplemental Information.

### RNA extraction, RNA-seq, data analysis, and gene expression analysis

The experimental details and the relevant methods are described in the Materials and Methods section of the Supplementary Information.

## Supplementary Information

## Acknowledgements

This work was supported by the King Abdullah University of Science and Technology and Agence Nationale de la Recherche (ANR-2010-JCJC-1608). The Institute of Plant Sciences Paris-Saclay benefits from the support of the LabEx Saclay Plant Sciences (ANR-10-LABX-0040-SPS).

### Author Contributions

AA Abulfaraj: conceptualization, data curation, formal analysis, validation, methodology, and writing—original draft, review, and editing.
K Mariappan: data curation, software, and formal analysis.
J Bigeard: conceptualization, methodology, and writing—review and editing.
P Manickam: visualization and methodology.
I Blilou: visualization, methodology, and writing—review and editing.
X Guo: methodology.
S Albabili: supervision and methodology.
D Pflieger: conceptualization and writing—review and editing.
H Hirt: conceptualization, supervision, funding acquisition, project administration, and writing—review and editing.
N Rayapuram: conceptualization, data curation, formal analysis, supervision, validation, investigation, methodology, writing—original draft, project administration, and writing—review and editing.

### Conflict of Interest Statement

The authors declare that they have no conflict of interest.

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
