## [Reviewer comments · Life Science Alliance]

The Arabidopsis homolog of human G3BP1 is a key regulator of stomatal and apoplastic immunity

Aala A. Abulfaraj, Kiruthiga Mariappan¹, Jean Bigeard, Prabhu Manickam, Ikram Blilou, Xiujie Guo, Salim Albabili, Delphine Pflieger, Heribert Hirt, Naganand Rayapuram
DOI: 10.26508/lsa.201800046

Review timeline:	Submission date:	6 March 2018
	1 st Revision Received:	6 March 2018
	1 st Editorial Decision:	4 April 2018
	2 nd Revision Received:	17 May 2018
	2 nd Editorial Decision:	17 May 2018
	3 rd Revision Received:	21 May 2018
	Accepted:	22 May 2018

Report:

(Note: Letters and reports are not edited. The original formatting of letters and referee reports may not be reflected in this compilation.)

REFeree REPORTS

Referee #1:

The ms. of Abulfaraj et al. describes AtG3BP1 knockdown alleles and over-expression lines to conclude that AtG3BP1 is a regulator of stomatal and apoplastic immunity. There are two issues with the results and conclusions of the ms. 1st, there is little if any mechanistic or functional data on AtG3BP1 as the cursory phosphorylation on Ser257 (identified in the Phosphat database), subcellular localization and transcriptome profiling are generally uninformative. The 2nd, related issue is that the Atg3bp1-1&2 knockdowns, despite apparently normal early vegetative growth, exhibit constitutive pathogen response (cpr) phenotypes (Figs. 3-6: elevated disease resistance, PR transcripts, H₂O₂, ROS & SA), implying genetically that AtG3BP1 negatively regulates stomatal immunity and post-invasive disease resistance. Similar cpr or autoimmune mutants of varied phenotypic severity often represent direct or indirect R protein guardees (Palma et al. 2010 PLoS Pathog; Bonardi et al. 2011 PNAS; Zhang et al. 2012 Cell Host Microbe; Lolle et al. 2017 Cell Host Microbe). Such a scenario for AtG3BP1 must be assessed by including crosses between Atg3bp1 knockdowns (see comment 6 below) and mutants affected in SA (biosynthetic mutants, NahG transgenics, and in R gene functions (eds1/pad4 and ndr1). Suppression of Atg3bp1 knockdown/out phenotypes in such double mutants/NahG would require further work to confirm that AtG3BP1 is directly or indirectly guarded, work well beyond the scope of the current manuscript.

Specific Comments

- 1- Fig. S1B appears incorrect as At5g47280, which in the phylogeny is the closest homolog to At5g48650 (AtG3BP1), encodes an unrelated R protein. The phylogeny should also include HsG3BP1 to substantiate a claim of 'closest homolog to HsG3BP1' (p. 6).
- 2- As GFP itself localizes similarly, a GFP western to verify accumulation of full-length AtG3BP1-GFP fusion is required for Fig. 2B.
- 3- Considering the phenotypic characterization, the plants were grown at a relatively high temperature (23°C) which may well mask cpr/autoimmune phenotypes as many R genes are less

active at higher temperatures. Also, only 4 week-old plants are shown such that phenotypes may not be fully developed yet. The authors should grow the plants at lower temperatures, include pictures at later developmental stages, and stain tissues with trypan blue to detect any microscopic cell death or lesions.

4- For the resistance assays, the authors need to show 0 hpi in spray assays to rule out different inoculum input as the source for the differences at 3hpi.

5- AtG3BP1 was chosen as phosphorylation of Ser257 is slightly increased following FLG22 treatment, presumably by a MAP kinase. Is there any effect of mutation of Ser257 (Ser-Ala or Glu) on the proposed function(s) of AtG3BP1?

6- Atg3bp1-1 presumably expresses an mRNA at ~10% WT levels of an mRNA encoding the NTF2 domain while Atg3bp1-2 (typoed in Fig. 1E&G) expresses appreciable levels of an almost full-length AtG3BP1. Is this the case, as the qPCR with upstream exon primers would not reveal this? In addition, it would be far cleaner to generate Atg3bp1 knockouts by CRISPER-Cas, and comparative knockouts of its real, closest homologs to assess redundancies.

Referee #2:

In the manuscript entitled "The Arabidopsis homolog of human G3BP1 is a key regulator of stomatal and apoplastic immunity", the authors identified a homologue of the human G3BP1 in Arabidopsis from a phosphoproteomic analysis of flg22-treated Arabidopsis plants. Atg3bp1 mutants display constitutively increased salicylic-dependent immune responses, which leads to enhanced resistance to *Pseudomonas syringae* infection. While the identification of AtG3BP1 as a negative regulator of immunity is of interest, the current manuscript seems premature - especially when considering publication in a journal such as EMBO J.

Major comments:

1. Although G3BP1 was first identified as a flg22-responsive phosphoprotein, the authors make no attempt to test the functional relevance of phosphorylation on Ser-257, and to identify which kinase(s) is/are responsible for this phosphorylation.

2. Similarly, G3BP1 is a conserved RNA-binding. So, is RNA-binding capacity required for the reported function of G3BP1 in plant immunity? To which transcripts does G3BP1 bind during immunity? The authors discuss the published roles of mammalian G3BP1 and Arabidopsis G3BP1-like in stress granules. Thus, is G3BP1 SG-localised?

3. The sub-cellular localization of G3BP1 is poorly investigated in the study. This should be analysed upon expression of the protein under the control of its native promoter, and it should be tested whether the epitope-tagged G3BP1 protein can complement the g3bp1-1 mutant. Also, sub-cellular localization of G3BP1 should be studied in leaf tissues, as these are the parts of the plant that are used for all experiments.

4. There are no experimental details provided for the phosphoproteomics experiments, which makes it difficult to assess whether G3BP1 is truly a MAMP-responsive phosphoprotein.

Minor comments:

5. The authors describe a family G3BP1-like proteins in Arabidopsis containing 17 members. Please discuss what criteria were used to define this family. Several members of the 'family' are actually NBS-LRR proteins (e.g. ADR1 discussed in the text), which do not seem to contain the NTF2 or RNA-binding domains of G3BP1 and seem to share very little sequence homology. Maybe it would also be helpful to include a multiple alignment in the supplemental figures. In addition, HsG3BP1 should also be included in the phylogenetic tree.

6. On a related note, no information is provided on how the phylogenetic tree and multiple alignment in Sup Fig1 were produced.

7. In the abstract, the authors claim there is no growth or developmental defects. This conclusion is too strong from only showing images of 4-week-old plants. Other parameters (e.g. biomass) could easily be quantified to observe if there are more subtle phenotypes, which would be expected from a

plant with some of the phenotypes shown by the mutant, such as constitutively reduced stomatal aperture.

8. There are several issues with Figure 2.

- In the materials and methods, the authors state that the seedlings are all 10-days-old. There is a clear age difference between panel I and II.
- The ATG3BP1 promoter does not appear to be active in leaves from the panel I and II, which seems to be in contrast to what observed in panels iii and iv. The stomata do not appear to have stronger staining in III; is panel IV representative?
- State which transgenic line was used. One has much higher expression and would be more likely to have overexpression artefacts.
- A better resolution would be required for publication.

9. The authors should test the genetic interaction between ATG3BP1 and key components of the SA signalling pathway e.g. crossing with *sid2* to determine if mutant phenotypes such as gene expression are abolished in an SA deficient background.

10. The authors should perform the stomatal assays following application of both exogenous coronatine and JA. The data suggests that there is a constitutive activation of SA signalling which cannot be suppressed by coronatine. Can this be overcome by application of higher concentrations of JA or coronatine?

11. The *Atg3bp1* alleles display higher immune responses but less callose deposition is observed in these mutants which seems counter-intuitive. The authors should discuss this point. Also, it is not correct to state that "AtGB3BP1...may not be important for some aspects of the late apoplastic MTI responses, notably callose deposition" given that the *Atg3bp1* mutants had less callose deposits upon *flg22* treatment. This actually demonstrates that AtGB3BP1 is important for this response.

12. In Figure 4, how do the authors explain that the *Atg3bp1-1* mutant show lower Pst DC300 titers at 3 hpi in panel D, while this is not the case on panel H; despite the fact that the legends indicate that the experimental conditions were similar?

13. The authors should consider renaming the gene considering Krapp et al. (2017) named a homolog of this gene 'G3BP-like protein (AtG3BP)', which will lead to confusion.

Referee #3:

This manuscript describes that *Arabidopsis* G3BP1 (At5g48650) is a negative regulator of plant immunity against the bacterial pathogen *Pseudomonas syringae*. Mutants deficient in this gene showed constitutive immune responses (enhanced ROS and SA and constitutive closure of stomata), which likely explain resistance phenotypes of this mutant.

The family G3BP genes have "almost" never been characterized in terms of plant immunity, and therefore genetic evidence provided in this manuscript is interesting and novel. However, this manuscript does not provide any mechanistic insight into how G3BP1 regulates immunity or how G3BP1 is regulated during immunity. Thus, this manuscript remains descriptive and to this reviewer does not significantly contribute to the field of plant immunity at the level of this journal. Many negative regulators in plant immunity whose mutants show constitutive immune responses have been characterized. Such mutants including the ones in this study show heightened ROS and SA signaling. Therefore, without showing a mechanism, this study joins in already existing collections of mutant reports for negative regulators of plant immunity whose functions remain obscure.

Constitutive defense mutants often exhibit growth defects. In contrast, mutants in this study do not appear to show such developmental phenotypes (at least the authors claim but evidence is weak, though). Thus, this observation could potentially be interesting if the authors show why mutants deficient in this gene show normal growth phenotype. Since defense and growth tradeoff is well described and a problem in agriculture, such mechanistic insight would attract large audiences.

Besides general significance, there are several issues (please see below in detail) that the authors

should consider to improve this manuscript. In particular, the phylogenetic tree (FigS1) seems not right.

In summary, this manuscript would be potentially interesting to a wide range of audiences but remains descriptive without a mechanistic insight. In addition, some conclusions are not justified by presented evidence and some descriptions are overstated. In addition, some experimental issues need to be solved.

(Major comments)

1. Regarding the phylogenetic tree of Arabidopsis G3BP family (FigS1). In the tree, the closest homolog (At5g47280) of the gene described in this study (At5g48650) is ADR1-L3, which belongs to NLR family like other ADR1 genes. G3BP and NLR family do not have similar structure or sequence, so this result appears to contain some error. I also double checked if At5g47280 and At5g48650 have similar sequences, but I failed to see a similarity. Related to this, in Discussion the authors discuss about At5g47280 (Page 16 "Interestingly, another closely related member of...including PR1"). I think that this paragraph has at least three errors. 1) At5g47280 does not seem to be related to G3BP. 2) At5g47280 is not ADR1 but is ADR1-L3. As far as I know, mutants of At5g47280 have not characterized in publication. 3) In the *adr1* mutant paper (Grant et al 2003 MPMI), they characterized activation tagging lines of ADR1 (more or less overexpression of ADR1). Furthermore, this paragraph is inconsistent with the statement in Introduction "so far no role of G3BP has been reported in plants". Related to this point, how did the authors identify Arabidopsis G3BP genes? This needs explanation.

2. Regarding *g3bp* showing disease resistance with constitutive defense up-regulation but without growth and developmental defects. Only evidence that the authors provided is Fig1G. Since this is a strong and important claim, the authors should test and quantify multiple aspects of growth and development (i.e. shoot fresh weight, reproduction...). Besides, to this reviewer, *g3bp1* mutants look smaller than WT plants. Expression of this gene in roots is high. Do mutants have root growth or developmental phenotypes?

3. Regarding G3BP1 regulating stomatal immunity via SA signaling. Page 13 "AtG3BP1... SA signaling"; "Collectively, these results confirm...in SA-mediated stomatal..." And Fig6 title are not supported by the provided data. *g3bp1* mutants show constitutive stomatal closure, which only associated with heightened SA signaling activity and accumulation. To conclude that G3BP1 regulates stomatal immunity via SA signaling, the authors need to test double mutants of *g3bp1* with SA-related mutants. Both SA biosynthesis and signaling mutants would be useful.

4. Fig2. Since the authors used leaves from mature plants for immunity assay (bacterial growth and stomatal assay), the authors should test expression of gene and fusion protein in mature leaves. Related to this, is gene expression affected by pathogen infection and MAMP treatments? In addition, the pictures in Fig2B are not clear. It is difficult to say in which compartments G3BP1 localizes.

5. Page 2 "insensitivity to bacterial coronatine...reopening" *g3bp1* mutants show constitutive closed stomatal phenotypes regardless of treatments (i.e. Fig6E). Therefore, insensitivity to bacterial coronatine mediated stomatal "reopening" is misleading. To this reviewer, stomata aperture of *g3bp* mutants does not change except after ABA treatment.

6. Regarding phosphorylation of G3BP1. Is phosphorylation important for function in immunity? It is rather surprising that the authors did not test by complementing mutant lines with phosphor-mimic and phosphor-deficient versions of G3BP1, since phosphorylation of G3BP1 in *flg22* response was the start of this study. These experiments will strengthen the quality of this study.

(Other comments)

- Please provide statistical analysis in Fig1E.

- Fig3E. Other defense responses are constitutively active or enhanced in *g3bp1* mutants, but only callose deposition is compromised. Is this due to this time point or general feature? Do mutants show enhanced callose deposition at different time points? The authors should at least discuss this.

- Fig5B. Cluster III looks also interesting. In mock samples, WT and mutant show similar expression patterns but those genes are highly induced in WT but not in mutant plants.

- Page 6. Since both g3bp1 mutants have an insertion at exons, to this reviewer, these mutants are not "transcriptionally knock-down mutants". Likely knockout mutants.
- As the authors discuss (Krapp et al 2017), this is not the first study showing implication of G3BP family proteins in plant immunity. Thus, Page3 "but so far no role of G3BP has been reported in plants" should be toned down.
- Page12 "Among these genes...SA-related genes" needs data.
- Transcriptome data. Since g3bp1 mutants show constitutive stomatal closure, differential transcriptome patterns in WT and mutants could be simply caused by different bacterial loads. To understand roles of G3BP1 in plant immunity, bacterial infiltration or MAMP treatments would have been much better. The authors can investigate by qRTPCR for some gene expression in WT and mutants in response to bacterial infiltration or MAMPs.

1st Revision – authors' response

6 March 2018

All the 3 reviewers pointed out the error in the inclusion of ADR1 (At5g47280) as an Arabidopsis G3BP homolog, on careful re-analysis of the data, we found that there was an error in the association of the Uniprot accession codes to the Arabidopsis AGI codes in the database, as the gene model considered is incorrect. To rule out the possible mistakes with the other not so obvious candidates, we looked up all the Arabidopsis G3BP homologs that we had reported in the manuscript and noticed that Uniprot had given different AGI codes to splice variants of the same gene. To rectify this error, we revised the number of Arabidopsis G3BPs genes to be only 8 in total after cross verification with the Arabidopsis TAIR database. The revised version of the manuscript submitted now incorporates all the corrections in the text as well as in the respective figures and expression profiles.

Below are the specific changes brought about in the manuscript.

Page 3, Line 21 - 22 : We have changed the sentence from "Here, we identify the Arabidopsis G3BP1 homolog that belongs to a family of 16 proteins and functions in plant immunity." to "Here, we identify the Arabidopsis G3BP1 homolog that belongs to a family of 8 proteins and functions in plant immunity."

Page 6, Line 1 – 3 : We have changed the sentence from "The Arabidopsis genome codes for 16 proteins, which satisfy these criteria. The 16 Arabidopsis G3BPs are named AtG3BP1 to AtG3BP16 as indicated in (Figure S1A)." to "The Arabidopsis genome codes for 8 proteins, which satisfy these criteria. The 8 Arabidopsis G3BPs are named AtG3BP1 to AtG3BP8 as indicated in (Figure S1A)."

Figure S1A has been updated to show only 8 G3BP proteins now.

Figure 1A now shows the tree with only 8 Arabidopsis G3BPs and the Human G3BP1.

Page 13, Line 20 – 22 : We have changed the sentence from "AtG3BP1 is a member of a family of 16 proteins in Arabidopsis that are orthologs to the human G3BP gene family." to "AtG3BP1 is a member of a family of 8 proteins in Arabidopsis that are orthologs to the human G3BP gene family."

Figure S4 has now been updated with the expression for the 8 G3BPs in place of the 16 G3BPs.

1st Editorial Decision

4 April 2018

Thank you for submitting your manuscript entitled "The Arabidopsis homolog of human G3BP1 is a key regulator of stomatal and apoplastic immunity" to Life Science Alliance. Your manuscript was previously reviewed at another journal, and the reviewer reports were confidentially transferred to Life Science Alliance with your permission. I had asked you prior to your submission to provide a point-by-point response and to address the following concerns of the previous round of review:

- Referee #1: specific comments

- Referee #2: points 3 and 4 and minor points 5-8.
- Referee #3: points 1 and 2, and the concerns in (other comments)

Your revised version has been seen by one of the original reviewers again, and I am afraid that this reviewer thinks that the data provided are not sufficiently addressing the concerns I asked you to address. Specifically, the sub-cellular localization of AtG3BP1 (referee #2, point 3) is not satisfactorily documented, and the anti-GFP blot for the overexpression line should be provided in full (referee #1, specific comment). Furthermore, the reviewer notes that the text could benefit from another revision.

I would thus like to ask you to provide a final revision, addressing these points. I should stress that we usually only allow a single round of revision.

Thank you for this interesting contribution to Life Science Alliance. We are looking forward to receiving your revised manuscript.

2nd Revision – authors' response

17 May 2018

Reviewer #1 (Comments to the Authors (Required)):

This new manuscript submitted is improved, mostly because it is now more focused on the genetic role of AtG3BP1 as a negative regulator of immunity against *Pseudomonas syringae*.

I however still feel that the sub-cellular localization of AtG3BP1 is poorly documented, as the Figure 2C would need to be at higher magnification to allow any meaningful conclusion. Also, while I appreciate the efforts done to provide the requested anti-GFP western blot for the line OE3, this analysis is not really useful without showing the full blot, as it is important to check whether the protein is cleaved given that free GFP will display the reported nucleo-cytoplasmic localization.

We have now thoroughly investigated the sub-cellular localization of AtG3BP1 in the stable Arabidopsis overexpressor lines as well as transiently in N. benthamiana leaf epidermal cells. We have modified Figure 2 with better images with higher resolution to clearly make a conclusion about the localization of AtG3BP1. In addition, we also carried out the western blot analysis and now we show the full blot to confirm that the GFP tag is not cleaved from the protein.

In the Introduction and Discussion, a number of references used to refer to MTI outputs are not always the original ones, or there is a lack of consistency in the citation of primary research articles versus review articles. My suggestion would be therefore to cite simply a recent comprehensive review on the topic, such as Yu et al., Annu. Rev. Phytopathol. 2017 (doi: 10.1146/annurev-phyto-080516-035649). Also, when referring to specific proteins (e.g. MPK3, MPK4, etc...), specify that these correspond to proteins known in *Arabidopsis thaliana*.

As per the suggestion by the reviewer, we have now cited the recent comprehensive review by Yu et al., and tried to make the citation more consistent.

Regarding the reference to specific proteins (eg., MPK3, MPK4 etc.), we have now changed the text as follows –

In Arabidopsis thaliana, a pathogen attack or treatment with a microbe-associated molecular pattern (MAMP) such as flg22 (the conserved 22-amino-acid peptide derived from Pseudomonas aeruginosa flagellin), leads to the recognition of the MAMP by the pattern recognition receptor (PRR) that activates MAMP-triggered immunity (MTI). Receptor-mediated recognition of MAMP leads to the induction of a battery of defense responses that include production of reactive oxygen species (ROS) (Torres, Jones et al., 2006), an increase in intracellular calcium concentration (Liese & Romeis, 2013), and activation of four known Arabidopsis mitogen-activated protein kinases namely MPK3, MPK4, MPK6 and MPK11 (Frei dit Frey, Garcia et al., 2014, Ranf, Eschen-Lippold et al., 2011, Zipfel, Kunze et al., 2006).

In the Introduction (p. 5), when summarizing the literature on auto-immune mutants and the phenotypes conferred by SA imbalance, it is not necessarily correct to simply refer to "pathogens", as the phenotypes are often opposite when considering hemi-/biotrophs versus necrotrophs.

We thank the reviewer for having brought this to our notice. We went back to literature and have now changed the sentence to read as follows –

Mutations in SA accumulation or signaling genes and Arabidopsis transgenic plants with bacterial gene nahG that encodes SA hydrolase were compromised in defense against P. syringae pv. Tabaci, Phytophthora parasitica or Cercospora nicotianae.

P. 7: the title "AtG3BP1 alters classical MTI responses" should rather be "Loss of AtG3BP1 alters classical MTI responses".

We have now changed the title of this section to read "Loss of AtG3BP1 alters classical MTI responses" as suggested by the reviewer

It would be better to express bacterial numbers as 'log₁₀ cfu/cm²' as this is the most commonly-used unit and thus would allow easier comparison with other publications.

We re-analyzed the data and have now modified and indicated the bacterial numbers as log₁₀ cfu/cm² in Figures 3A, Figure 3B, Figure 4D, Figure 4G and Figure 4H.

Discussion, p. 14: it is odd to write "...confirmed the role of AtG3BP1 in plant innate immunity", as it suggests that this role had already been reported before.

As per the suggestion by the reviewer, we have now changed the sentence to read as follows –

"The enhanced disease resistance to Pst observed in two independent Atg3bp1 loss of function mutant lines indicates a role for AtG3BP1 in plant innate immunity."

Given that the authors report here the family to which AtG3BP1 belongs to, it would be good, for clarity, to refer to the previously studied At5g43960 as AtG3BP7 when first referred to in the text.

We have now made this modification in the discussion part of the manuscript as we propose the nomenclature for these genes in the first part of the results section. We decided to call it AtG3BP-like (At5G43960) in the introduction where we first refer to this gene.

In the Discussion, it would be great if the authors could speculate on why altered AtG3BP1 expression affects ROS production, which is a very early MTI responses occurring within a few minutes. Also, given that AtG3BP1 has homology to a RNA-binding protein, it would be good to discuss at least how this protein could regulate immunity mechanistically.

We have now added the following paragraph in the discussion to speculate how AtG3BP1 might regulate stress and immunity in plants:

RNA-binding proteins especially those involved in response to stress have been shown to regulate gene expression at different stages, from transcription to post-translational levels (Mastrangelo, Marone et al., 2012). Post-transcriptional regulations are important processes in eukaryotes that enable them to adapt to changes in their environment. Human G3BP1 is located in stress granules, which are sites of mRNA storage during stress responses that are involved in regulating mRNA stability and translatability (Anderson & Kedersha, 2008). In humans, SGs are highly dynamic entities that recruit many signaling proteins in a transient manner until the cells adapt to stress or die. Hence, beyond their role as mRNA triage centers, SGs are proposed to constitute RNA-centric signaling hubs analogous to classical multiprotein signaling domains such as transmembrane receptor complexes which communicate a 'state of emergency' (Kedersha et al., 2013). A good example for this as signaling hub is the shown by HsG3BP1 which functions in regulating double-stranded RNA-dependent protein kinase activation (Reineke et al., 2015). Given the highly conserved nature of eukaryotic G3BPs, it is tempting to speculate that plant G3BP1 might function

in an analogous fashion during stress. This might also explain the fact, that Atg3bp1 mutants showed enhanced ROS accumulation in response to flg22, which would suggest that AtG3BP1 functions upstream of ROS production. In this context, it should also be noted that ROS production does not only act as an anti-microbial agent in MTI, but also as secondary messenger that triggers downstream defense responses, including stomatal closure and up-regulation of MTI marker genes (Kadota et al., 2014, Melotto et al., 2006, Yu et al., 2017).

Interestingly, whereas we identified eight G3BPs in Arabidopsis, the human genome only encodes for two HsG3BPs. Expression analysis of the Arabidopsis genes shows however that these AtG3BPs are expressed in a highly tissue- and developmental-specific fashion (Figure S4), suggesting that the different genes might perform similar functions in different cellular contexts.

2nd Editorial Decision

17 May 2018

Thank you for submitting your revised manuscript entitled "The Arabidopsis homolog of human G3BP1 is a key regulator of stomatal and apoplastic immunity".

I appreciate the introduced changes, and I am happy to accept your manuscript in principle for publication in Life Science Alliance.
